# Lithium in Portuguese Bottled Natural Mineral Waters—Potential for Health Benefits?

**DOI:** 10.3390/ijerph17228369

**Published:** 2020-11-12

**Authors:** Maria Orquídia Neves, José Marques, Hans G.M. Eggenkamp

**Affiliations:** 1Department of Civil Engineering, Architecture and Georesources, CERENA (Centro de Recursos Naturais e Ambiente), Instituto Superior Técnico, University of Lisbon, 1049-001 Lisbon, Portugal; jose.marques@tecnico.ulisboa.pt; 2Department of Petrology and Mineral Resources, Eberhard-Karls University of Tübingen, Schnarrenbergstraße 94-96, 72076 Tübingen, Germany; hans@eggenkamp.info

**Keywords:** lithium intake, natural mineral water, health benefits, public health

## Abstract

There is increasing epidemiologic and experimental evidence that lithium (Li) exhibits significant health benefits, even at concentrations lower than the therapeutic oral doses prescribed as treatment for mental disorders. The aim of this study is to determine the content of Li in 18 brands of bottled natural mineral waters that are available on the Portuguese market and from which the sources are found within the Portuguese territory, to provide data for Li intake from drinking water. Analyses of Li were performed by inductively coupled plasma-mass spectrometry. The results indicate highly different Li concentrations in natural mineral waters: one group with low Li concentrations (up to 11 µg Li/L) and a second group with Li concentrations higher than 100 µg/L. The highest Li concentrations (>1500 µg Li/L) were observed in the highly mineralized Na-HCO_3_ type waters that are naturally carbonated (>250 mg/L free CO_2_). As a highly bioavailable source for Li dietary intake these natural mineral waters have potential for Li health benefits but should be consumed in a controlled manner due to its Na and F^−^ contents. The consumption of as little as 0.25 L/day of Vidago natural mineral water (2220 µg Li/L), can contribute up to 50% of the proposed daily requirement of 1 mg Li/day for an adult (70 kg body weight). In future, Li epidemiological studies that concern the potential Li effect or health benefits from Li in drinking water should consider not only the Li intake from tap water but also intake from natural mineral water that is consumed in order to adjust the Li intake of the subjects.

## 1. Introduction

One of the challenges of the present century is the improvement of human health and to prevent the spreading of diseases. This specifically also applies to mental disorders that occur in all regions and cultures of the world. The most prevalent of these being depression and anxiety, which are estimated to affect nearly one in ten people on the planet. At its worst, depression can lead to suicide [1].

Lithium is the gold standard treatment for several psycho-neurological diseases (e.g., as bipolar disorders). The relationship between Li and health has been shown over time, since the time of the Roman Empire but the clinical history of lithium only started in the mid-19th century when it was used to treat gout, that proved to be ineffective. Its use in the treatment of psycho-neurological began in 1948, by John Cade [2] in Australia. It is administered essentially as carbonate (Li_2_ CO_3_) and at therapeutic doses within the limits of 600 to 1200 mg/day (113–226 mg Li/day [3]). Due to the toxicity of Li there is a rather narrow therapeutic window (between 0.6 and 1.2 mmol/L blood serum) for Li medication, which must be continuously monitored. Lithium is known to interact with neurotransmitters and receptors in the human brain, increasing serotonin levels and reducing brain production of norepinephrine. The mechanisms under which Li acts neurologically have yet to be fully understood, although several hypotheses exist [4].

In the human body the average (total) Li quantity is approximately 7 mg [5] and it is found in various organs and tissues. Schrauzer [6] reported that Li appears to have an important role in fetal development, considering the relatively high Li content within embryos during the early pregnancy. Post-mortem human studies revealed that the cerebellum, cerebrum, and the kidneys retain more Li than other organs [6]. Although, Li was not yet officially recognized as an essential element and no recommended dietary allowance was proposed, in 2002 Schrauzer [6] indicated for a 70-kg adult a provisional daily requirement of 1 mg Li/day (14.3 µg/kg body weigh).

All the essential elements and those considered beneficial must be provided by the diet or nutritional supplements.

Environmental Li exposure and population diet intake can vary greatly from region to region. The available data of daily Li intake point low to average doses in Belgium (8.6 µg/day) [7], the United Kingdom (16 µg/day) [8], France (48.2 µg/day) [9], Hanoi (Vietnam) (36 µg/day) [10] and New Zealand (20–29 µg/day) [11]. Depending on different Li content in food and beverages and to different ingestion habits, the intake could be significantly higher as in Canary Islands (3.6 mg/day [12]) and vary over a wide range [6]. Based on literature data, some grains and vegetables are the primary sources of Li (0.5–3.4 mg Li/kg) as compared to dairy products (0.5 mg/kg) and meat (0.012 mg/kg) [13]. To meet the nutritional demand for Li, Goldstein and Mascitelli [14] suggested that cereal grain products should be fortified with Li or it be added to dietary supplements. Mleczek et al. [15] also investigate mushrooms Li fortification, as food or alternative medicine in various cultures but further studies are necessary to investigate the safety implications of these Li-enriched food items.

Not only solid food, which is the major source of mineral nutrients in the human diet but also drinking water can contribute with variable amounts to the total intake. The role of non-alcoholic beverages was reported in a French diet survey where it was observed that important contributions to Li intake were water (35% for adults), followed by coffee (17%) and other hot beverages (14%) [9]. Lithiated beverages were common in the beginning of the twentieth century, as they were believed to mediate health benefits. One of the most popular soft drinks in the world was launched in 1929; the “*Lithiated Lemon Soda*” that was supplemented with 5 mg Li (as Li citrate/L) until 1948 [16], when it was banned by the government. It was believed to cure alcohol-induced hangover symptoms, make people more energetic and give lust for life and on the top of that shinier hair and brighter eyes [17]. In fact, it is still on the market but since 1936 its name changed to 7UP. In 1949, John Cade discovered that higher Li concentrations were toxic. Nowadays, according Seidel et al. [16] 7UP only contains 1.4 µg Li/L.

In recent years, there have been ecological studies on aggregate data that suggest that long-term intake of low Li concentrations, such as occurring in public drinking water (tap water), may also promote mental health benefits for the general population. This research found that higher concentrations of Li in the tap water are associated with lower suicide mortality rates. These results were observed in Texas (1–160 µg Li/L [18]), Japan (1–60 µg Li/L [19]), Austria (<3–1300 µg Li/L [20]), Greece (0.1–121 µg Li/L [21]) and Lithuania (0.5–35 µg Li/L [22]). This inverse association was found with or without adjustment for additional confounding factors such as the socioeconomic factors that are closely related to suicide. However, in the east of England where Li concentrations in tap water are between 0.1 and 21 µg/L [23], in Italy (0.11–60.8 µg Li/L [24]), Denmark (0.6–30.7 µg Li/L [25]) and in Portugal (<1–191 µg Li/L [26]) the association that high Li concentration in drinking water may protect against suicide was not well supported.

Lithium has also been considered as a possible therapeutic agent for treating chronic neurodegenerative diseases such as Alzheimer’s, Parkinson’s, and Huntington’s diseases [27]. A Li dose of 300 µg/day has been reported to stabilize cognitive impairment in patients with Alzheimer’s disease although the underlying molecular mechanisms have not yet been fully understood [28]. Furthermore, there is experimental evidences that Li may have positive effects on bone health [29] and muscle function [30].

From these findings it has been suggested that Li should be added to public drinking water supplies to improve the mental health of the general population, although this would be premature and raises ethical concerns [31] and further research on this subject is necessary.

Lithium doses used for mental health treatment are considerably higher than those obtained from daily exposure to Li in tap water. This raises questions regarding whether (an increased) daily intake of Li from tap water can reduce the risk of suicide or otherwise be beneficial towards the mental health of the population. As reported above, tap water is not the only liquid dietary source of Li and the earlier discussed studies did not take it into account.

Several studies have shown that various bottled waters are rich in Li; the highest values reported (9860 and 5450 µg Li/L) were from bottled waters from Slovakia [32] and Armenia (Hankavan-Lithia: 5.45 mg Li/L). Mineral waters such as Vichy Catalan (1.3 mg Li/L) and Evian (6.6 µg Li/L) were initially also promoted as Li waters based on their Li content [33].

Bottled water plays a more and more important role in daily life. The worldwide bottled water consumption is characterized by a significant growth over the last decade. Its consumption is still increasing [34], especially in developed countries, even though tap water quality is good and several orders of magnitudes less expensive than bottled water. Although all bottled waters might look the same, in fact each natural mineral or spring water has its own distinctive taste, a unique set of properties and a specific chemical composition at the source from where it is extracted, that reflects the geological characteristics of the region and water-rock interactions occurring at depth. Moreover, the defining characteristics of naturally sourced waters are reflected in their protected origin status and are guaranteed by strict European Union (EU) legislation governing the extraction and packaging of the product [35].

European and national legislation distinguish three categories of waters: natural mineral water, spring water and drinking water.

The EU has laid down specific rules for natural mineral and spring waters, which clearly set them apart from drinking water, more commonly known as tap water [36]. Lithium is one of the elements for which no potable water standards are defined in Europe. In Australia Li is listed as a pollutant that causes environmental harm and it is limited to 2.5 mg/L for general irrigation and to a limit of 0.075 g/L for the irrigation of citrus cultures, respectively [37].

Bottled water in the EU is predominantly made up of the natural mineral water category [35]. The bottling and commercialization of natural mineral waters first began in Europe in the mid-16th century, with the mineral waters from Spa in Belgium, Vichy in France, Ferrarelle in Italy, and Apollinaris in Germany.

In 2016, natural mineral water accounted for 83% of EU bottled water retail, with spring water accounting for 14% [34].

In Portugal, like in other countries, natural mineral and spring waters have always aroused great interest due to their exceptional quality, diversity, and health-friendly effects. In 2018, the Portuguese per capita consumption of bottled waters was 134 L/year, which is the 7th highest European Union average consumption (EU average is 119 L/year) [35].

Each natural mineral water in Europe must receive official recognition from the State’s competent authority. In the list of natural mineral waters brands recognized by the EU 27 Member States, Portugal accounts for 22 brands [38].

According to Portuguese Legislation [39] natural mineral waters (the subject of this study) are bacteriologically pure waters, of underground circulation, with stable physico-chemical characteristics at the source within the range of natural fluctuations and which may result in possible therapeutic properties or favorable health effects. Spring waters are also natural waters of underground circulation, bacteriologically pure, which do not have the characteristics necessary for qualification as natural mineral waters, provided that at the source they are suitable for drinking.

The present study aims to quantify the Li concentration in Portuguese marketed bottled natural mineral waters to identify if they may represent a significant source of dietary Li intake. Due to its biological impact, it becomes more and more important to understand the Li content in drinking water obtained from different sources, particularly in bottled natural mineral waters, especially in big cities. These data will be very helpful for evaluating the future Li intake via drinking water and diet of Portuguese epidemiological studies related with Li health effects on the population or at an individual level, also contributing to the development of a Li food data base.

## 2. Materials and Methods

### 2.1. Sample Collection

The eighteen brands of bottled natural mineral waters characterized in this study were selected from the list of the bottled natural mineral waters recognized by Portugal, updated in September 2019 [39]. On this list, the brands Monchique and Chic are, according their label, the same mineral water but commercialized in different packages. Also, the mineral waters Pedras Salgadas and Pedras Levíssima only differ in their CO_2_ content [40].

All studied natural mineral waters were purchased in supermarkets and local shops. Regarding the type of packaging, nine of the natural mineral waters were bottled in polyethylene terephthalate (PET) and ten in glass bottles. It is not expected that the content of Li and other trace element under study in these mineral waters can be affected by bottled material leaching. According some experiments [32] this becomes problematic for Sb in PET bottles and for Pb, Cr and Ce in glass bottles at acid pH but not for Li.

These natural mineral waters have their catchment area on the Portugal mainland. Only the brand Magnificat issues at the volcanic island of São Miguel, Azores Archipelago (Figure 1).

### 2.2. Sample and Data Analysis

The pH value of the samples was measured with a glass electrode connected to a WTW pH 325-meter, previously calibrated against buffer solutions at pH 4.0, pH 7.0 and 10.0 (Merck), with an accuracy of the pH measurement of about ±0.05 pH units. Electric conductivity corrected to a temperature of 25 °C was measured using a WTW Cond 330i probe (WTW, Weilheim, Germany), previously calibrated with a 0.01 M KCl standard solution (WTW).

The anions, fluoride, chloride, nitrate, and sulphate were analyzed in non-acidified samples at the Laboratory of Mineralogy and Petrology of the Instituto Superior Técnico (LAMPIST, Lisboa, Portugal) by ion chromatography. A Thermo Scientific^TM^ Dionex^TM^ ICS-900 (Dionex, Sunnyvale, CA, USA) with auto-sampler equipped with a conductivity detector, an IonPac AS22 column and a self-regenerating suppressor using a sodium carbonate–sodium bicarbonate eluent was used for the analyses. Standard solutions (Merck) and ultrapure deionized water (resistivity 18.2 MΩ.cm at 25 °C) produced in a Direct-Q^®^3 water purification system (Merck Millipore) were used to prepare calibration standards. A multi ion anion IC standard solution (Alfa Aesar Specure) was used every batch of samples as reference and recalibration was performed if the average of triplicate measurements deviated by more than 10%.

The alkalinity was determined by volumetric titration on unfiltered and unacidified samples (50 mL) with a 0.02 N HCL solution, using an automatic Metrohm titrator (titration end point pH 8.3 to determine OH- and/or CO_2_^3−^_,_ followed to titration end point pH 4.5 to determine HCO_3_^−^ concentration, according to Standard Method 2.320B [42]).

The dry residue (DR) content, was obtained from evaporation and drying the water sample at 180 °C.

The cations were analyzed at Activation Laboratories, Ltd. (Actlabs) (Vancouver, BC, Canada), an accredited Laboratory, by inductively coupled plasma-mass spectrometry using a Thermo iCAP Q, after the samples had been acidified with concentrated HNO_3_ (≥65%) to a pH < 2. The Detection and Quantification Limits (LOD/LOQ) in μg/L are as follow: Li (0.02/0.06), Na (2/7), K (1/5), Ca (20/70), Mg (2/6), Mn (0.1/0.3), Rb (0.002/0.006) and Cs (0.001/0.003). Quality controls were achieved according to the Laboratory standards methods and quality assurance and protocols. The standards NIST 1643e and SLRS was used by Actlabs to check the validity and reproducibility of the results.

All samples were analyzed without filtration to represent the water consumed as it is in the bottle.

The charge-balance errors, based on the percentage difference between the total positive charge and the total negative charge (mEq), was below 10% for each sample.

Pearson’s correlation coefficient and linear regression with a confidence interval of 95% were calculated with TIBCO® Data Science—Statistica^®^ (Palo Alto, CA, USA) software (version 13.5.017) and Piper diagram projections with RockWorks17 software.

## 3. Results and Discussion

The natural mineral waters discussed in this study are groundwaters abstracted from boreholes and bottled directly at the source. Their distribution across the Portuguese mainland is uneven, with a greater concentration in northern part of the country (Figure 1), mainly caused by the (i) geomorphologic and climatic conditions (higher mountains, colder climate, more rainfall/recharge), (ii) structural characteristics such as the prevalence of deep faults responsible for meteoric waters infiltration at deep and natural mineral waters up flow to the surface as springs and (iii) geological signatures (more fractured and permeable rocks promoting water-rock interaction at depth and developing different water geochemical characteristics (e.g., Reference [43]).

The main characteristics of the studied Portuguese bottled natural mineral waters are presented in Table 1.

The natural mineral waters are commercialized with or without CO_2_ gas (carbonated or still water, respectively). In some of the sources, the dissolved CO_2_ can be present due to natural geological processes. If bottled as such it must be labelled as “naturally carbonated natural mineral water” (Table 1: NC waters). For example, in the case of Vidago and Pedras Salgadas waters, following [44] and references therein, their δ^13^ C_CO2_ values vary between −7.2 and −5.1‰ vs. V-PDB and CO_2_/^3^He ratios range from 1 × 10^8^ to 1 × 10^9^, indicating a deep (upper mantle) source for the CO_2_. It may also be possible to capture the natural source of CO_2_ and re-inject it into the water prior to bottling or added it artificially, being, in the second case, described as “artificially carbonated natural mineral water” (Table 1: AC waters). If the waters are subjected to gasification processes, this must be indicated on the label.

Considering the major ions present (expressed as percentage of the total mEq/L), the natural mineral waters under study are mainly of the Na-HCO_3_ type, follow by the Na-Cl type (Salutis, Vitalis, Caldas de Penacova and Luso) and Ca-HCO_3_ type (Castello and Melgaço) (Table 1 and Figure 2).

The natural mineral waters studied present a large range of Li concentrations. It ranges from less than 1 to 2210 μg/L and two groups can be recognized from the dataset: one group with low Li content (up to 11 μg/L) that represents 55.5% of the natural mineral water samples and a second group with higher Li contents (173 to 2210 μg/L).

### 3.1. Bottled Natural Mineral Waters with Low Li Content

The group with low Li samples ascribed to natural mineral waters with Li ranging between <1 and 11 μg/L) are also waters with very low (DR < 50 mg/L) or low dissolved salts (50 < DR < 500 mg/L). An exception is Vimeiro, with 2291 mg/L, due to water circulation through evaporite (with halite and gypsum) and carbonate rocks that occur at the contact of diapiric structures [45]. The Li concentrations measured in this group compare to Li concentrations observed in public drinking waters from 54 Portuguese municipalities [26]. According to Neves et al. [46], 75% of the water samples studied by Oliveira et al. [26] show concentrations below 10 μg Li/L, with a median of 4 μg Li/L. This is also lower than the median of 14.9 μg/L detected for Li, in 1785 samples of bottled waters collected in 38 European countries and analyzed for the European Groundwater Geochemistry Atlas [32]. In comparison to Li data available from German beverages such as wine (11.6 ± 1.97 μg/L), beer (8.5 ± 0.77 μg/L), soft and energy drinks (10.2 ± 2.95 μg/L) as reported by Reference [16], these Portuguese natural mineral waters can be considered Li-poor food items. The contribution of these natural mineral waters to the dietary Li supply will not be significantly different from the contribution of tap water.

Little is known on the effects of dietary Li on the Li status in the human body that is estimated either from the concentration in blood (plasma or serum) or from urinary Li excretion. Like it is for sodium, Li homeostasis is adaptively regulated by the kidney and Li is mainly reabsorbed in the proximal tubule. Under normal conditions, approximately 80% of Li is reabsorbed by renal tubes [47]. Excretion of Li occurs within 24 h after its oral intake and is facilitated by the kidneys. A small extent (2–3%) it is also excreted with feces and sweat [48].

Considering that the amount of Li taken by drinking water or food is probably reflected in serum or urinary Li levels, it will be necessary measure such levels, particularly in individuals that do not receive Li therapy.

In a study performed by Bochud et al. [49] both serum and urinary lithium concentrations were measured in Belgians and South Africa participants and in the tap water consumed by them (10 µg Li/L and 0.21 µg Li/L, respectively). Their results showed that the 24-h urinary lithium excretion was higher and more dispersed in the Belgians than in the South African participants (8.2 ± 5.6 and 3.1 ± 4.1 μmol per 24 h) but serum lithium levels were almost identical (0.31 ± 0.16 and 0.32 ± 0.21 μmol Li/L). These observations suggested that serum lithium is tightly regulated even when there are large variations in Li dietary intake from natural sources.

No increase in serum Li concentration was also observed by Seidel et al. [50] in the group that received low Li mineral water (1.7 µg/L) and they reported that the 24-h urinary Li excretion exceed the total uptake. At very low dietary intake, filtered Li is not fully reabsorbed [51] and Seidel et al. [50] suggested that there could be a minimum dietary need for Li to ensure a positive Li balance.

Concerning the protective effect of low exposure of Li from drinking water, future epidemiologic studies are required. The only individual-level cohort study carried out, on the Danish population for the period 1991–2012 [25], did not find any association for Li levels up to 31 μg/L in drinking water. However, information is still lacking regarding the quantity and/or duration of low Li exposure that is necessary to be achieved for relating Li either with anti-suicide effects or to reduced aggressivity and impulsivity, both associated with an increased risk of suicide [52].

### 3.2. Bottled Natural Mineral Waters with Higher Li Content

The group of the natural mineral waters with higher Li contents (173 to 2210 μg/L) are mostly of the Na-HCO_3_ water type, except in the case of Melgaço that is of the Ca-HCO_3_ type (ascribed to granodioritic rocks [53]). The highest Li content is measured in Vidago (Table 1). These waters are mainly exploited from Hercynian granitic rocks in the north of Portugal, from within the Minho and Trás-os-Montes regions, in the Geotectonic unit Galiza-Trás-os-Montes Zone (Figure 1). The catchment areas of these Li-rich waters is well correlated with regional fault systems, such as the “Penacova-Régua-Verin Fault”(Campilho, Vidago and Pedras Salgadas), the “Vilariça Fault” (Frize and Bem-Saúde) and the “River Minho Fault” (Melgaço) (Figure 1), since they normally provide the best conditions for the rising of fluids from deep crustal zones [43,44,45,53,54]. With exception of Campilho, all these Li-rich natural mineral waters are naturally carbonated, with free CO_2_ contents above 250 mg/L, identified on the bottle label as a “gasocarbonic” water. Carbonated waters were preferred by the consumers, as in addition to the slightly acidic taste, it stimulate the papillae tastes, favors digestion and especially if they are sodium carbonated water, they help to neutralize the acidity of the stomach [55].

Natural mineral waters with Li contents above 1500 μg/L present also high dissolved solids (DR > 1000 mg/L) and they are rich in sodium (Na > 400 mg/L), potassium (K > 27 mg/L) and magnesium (Mg > 10 mg/L). A good correlation was observed between Li and Na (*r* = 0.966, *p* < 0.05) and between Li and K (*r* = 0.976 *p* < 0.05), as also reported by Reference [50]. As their Na contents are higher than 200 mg/L they are classified as “water with sodium” [36] and so a regular consumption of these waters is not recommended for individuals that are on a low sodium diet.

Other elements that stand out in this Li rich group of bottled natural mineral waters, are fluorine (F^−^), rubidium (Rb) and cesium (Cs) (Table 1). Fluorine ranges from 1265 μg/L (Pedras Salgadas) to 4131 μg/L (Campilho), which according to Reference [36] can be also classified as “water with fluorine” (F^−^ > 1 mg/L). In this group the Vidago natural mineral water with only 25 μg F^−^/L, is the exception. According to Calado and Almeida [56], this anomalous F^−^ content, that does not result from the dissolution of fluorite, as it was supposed but has a deep genesis, related to the circulation of mineralizing fluids meso and/or infra-crustal origin. These fluids will be related to the lifting crustal phenomena (uplift) that mainly affect the north and center of the country [56].

The European Union (EU) Directive [57] defines a maximum admissible concentration of 5 mg F^−^/L in natural mineral waters and requires that F^−^ concentrations above 1.5 mg/L are indicated on the label, following the general EU Directive [58] for drinking water. Excessive F^−^ (>1.5 mg/L) incurs a risk of possible dental fluorosis, especially when the water is drunk regularly by children below the age of 7 and should be avoided also by adults.

Rubidium and Cs are also elements that are reported, together with Li, as a characteristic for some mineral waters related with water circulation through Hercynian granites. The maps of Rb (maximum 673 μg/L) and Cs (maximum 415 μg/L) in European bottled water [32] also shows up flow sites in northern Portugal and France (Massif Central) related to young granitic intrusions and complex type pegmatites of the LTC (Li-Ca-Ta) family.

It should be noted here that Li in water is present in aqueous solution in the form of hydrated Li^+^ ions. Being in solution, it may assimilate in the human body more easily as compared to solid food or with the salts as used in medication. If the beneficial effect of Li could be achieved at safer lower doses, increasing its dietary intake would offer an approach to the prevention of the incidence of mental disorders and a reduction in suicide attempts, aggressive behavior and conducted disorder as reported [18,59,60,61]. This way, it may be possible to reduce the amounts necessary to be administered and could also reduce side effects.

A recent study with healthy male volunteers [50] indicates that Li derived from medium to high Li mineral water (171 and 1724 μg/L, respectively), is highly bioavailable. The consumption of the mineral water with higher Li concentrations resulted in a peak serum content of up to 10–12 μmol Li/L which did not return to baseline levels within 24 h. Also, the total urinary excretion of Li was positively associated with Li uptake via mineral water. The data suggested that some minerals waters are an important and bioavailable Li source for human intake. If confirmed, these findings have public health relevance and emphasize the need for more data on Li concentrations in drinking water, as bottled natural mineral water, and their intake in a daily basis.

Among the Portuguese bottled natural mineral waters, Campilho, Frize, Pedras Salgadas, Bem-Saúde and Vidago waters, are the ones that can contribute more intensively to a significant Li absorption or even to reach the provisional daily adult intake (1 mg Li/day).

A consumption of 0.5 L of each of these waters may provide between 0.75 and 1.1 mg Li/day, assuring adequate Li intakes, especially people that are at risk of Li nutritional deficiencies. So, as a source of bioavailable Li, its amount and frequency of ingestion cannot be ignored in ecologic/epidemiologic Portuguese studies evaluating relations with intake of natural doses of Li via drinking water or diet and mental health benefits.

It must be realized that the studied Li mineral waters are also rich in other elements as F^−^ and Na, which can limit recommendations to be consumed regularly as source of Li for all individuals. For example, Vidago natural mineral water, due its lower F^−^ content could be indicated as one that can be used for this purpose but attention should be put to its Na^+^ concentration, as excessive Na in the human diet can harm the kidneys and acerbate high blood pressure that is associated with hypertension and coronary diseases in some individuals [62]. On the other hand, a higher Na content in the water can also modify the Li absorption process, as an increased of Na intake may increase the excretion of Li [48].

Considering a consumption of 0.5 L, Vidago natural mineral water may provide up to 315 mg Na/day, which is approximately 15.7% of the Daily Value (DV: 2 g Na/day) recommended by World Health Organization (WHO), for consumption. As a general guide, 5% DV or less of Na per serving is considered low and 20% DV or more is considered high [63].

However, Vidago natural mineral water also contains higher amounts of bicarbonate ion (HCO_3_^−^) instead of chloride (Cl^−^) as the anion associated with the Na^+^ cation. This is relevant because it is established that the effect of sodium in blood pressure depends on the corresponding anion; the blood pressure effect of sodium bicarbonate is much lower than that of equivalent amounts of sodium chloride [64]. A crossover, non-blinded study that evaluated 17 individuals ingesting 0.5 L/day of Pedras Salgadas and Vitalis natural mineral water (on Table 1) for 7 weeks, shows no effect on blood pressure values on normotensive individuals [64]. Another study conducted by Schorr et al. [65] also found that the ingestion of HCO_3_^−^ rich water (1.5 L/day) had hypotensive effects in and elderly population. However, this study was not replicate with hypertensive individuals, more prone to salt sensitivity. So, further research is necessary to improve knowledge on human body interactions with these anions and the benefits of these Li-rich natural mineral waters for our mental health.

Vidago natural mineral water is usually sold and consumed in 0.25 L bottles. The regular daily consumption of this natural mineral water volume can also provide a continuous supply of 550 µg Li per drink (half of the provisional Li daily intake) and Na with lesser health concerns, as it supplies 160 mg Na (8% DV) per serving. In that condition this mineral natural water will have potential to be regarded as an available natural nutritional Li supplement for suggested health benefits.

## 4. Conclusions

Depending on the chemical composition, natural mineral waters may significantly contribute to the recommended daily intake of minerals and provide us with a natural source of healthy hydration.

The health effects of Li in drinking water, both tap waters and bottled waters is not fully understood yet but there are indications that natural moderately high Li contents may be beneficial to the mental health situation of the population. In the present study the Li content from a set of bottled natural mineral waters from Portugal was evaluated. Based in the Li content two sets of natural mineral waters could be recognized. A set with low Li concentrations (up to 11 μg/L) that will not have any different effects on the dietary Li supply as compared to the contribution of tap water to the dietary contribution of Li and a second set with higher Li concentrations (173 to 2210 μg/L). The natural mineral water with higher Li contents (>1500 μg/L) is highly mineralized, mostly Na-HCO_3_ type waters and naturally carbonated (CO_2_-rich waters with > 250 mg/L free CO_2_). These Li-rich natural mineral waters can be a source of bioavailable Li and its consumption cannot be ignored in studies evaluating the intake of natural doses of Li via drinking water or diet. It thus is important to take into consideration for studies towards Li health effects or benefits for the population or at an individual level. Among the studied bottled natural mineral waters, the consumption of 0.25 L/day of Vidago natural mineral water can contribute significantly to reach the proposed provisional Li daily intake.

It should be noted that the dose of Li ingested through bottled natural mineral water is significantly less than the recommended doses for therapeutic purposes and for that reason these waters can be also regarded as a natural nutritional supplement.

Lithium’s interaction in human biochemistry is complex and should be a subject for continuous research to demonstrate the possible clinical effects of natural low-dose Li intake on mental health of the public.

## Figures and Tables

**Figure 1 ijerph-17-08369-f001:**
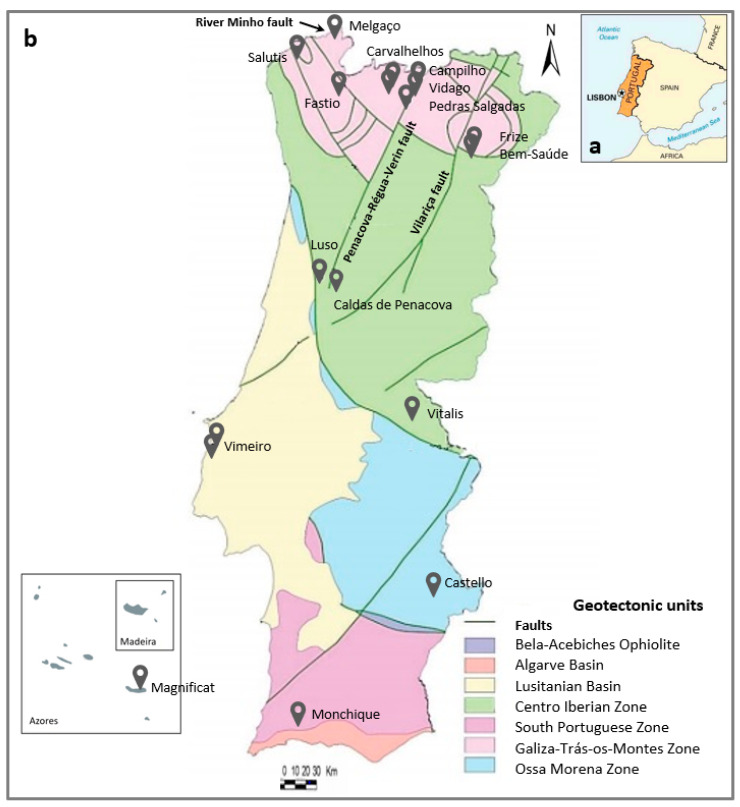
(**a**) Portugal geographic location; (**b**) Distribution of the catchment location of the studied Portuguese bottled natural mineral waters (adapted from Reference [41]).

**Figure 2 ijerph-17-08369-f002:**
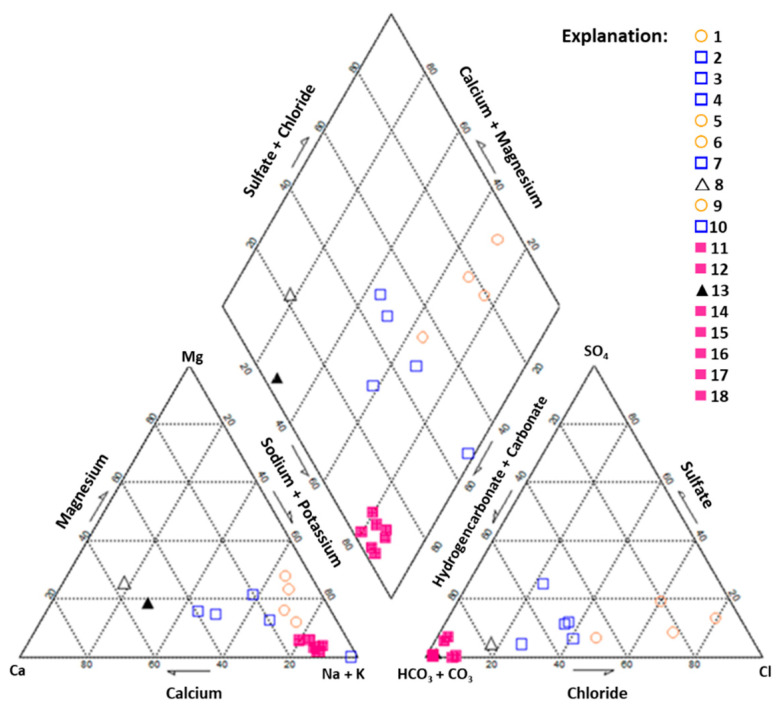
Piper Diagram showing the chemical composition of the studied Portuguese natural mineral waters (water type legend: Ca-HCO_3_ (triangles); Na-Cl (circles), Na-HCO_3_ (squares); low and high Li content (open and closed symbols, respectively); sample identification (number) as indicated in Table 1).

**Table 1 ijerph-17-08369-t001:** Characteristics of the studied Portuguese bottled natural mineral waters.

Mineral Water Brand		Type	pH	EC	DR	F^−^	Cl^−^	HCO_3_^−^	NO_3_^−^	SO_4_^2−^	Na^+^	K^+^	Ca^2+^	Mg^2+^	Mn^2+^	Li^+^	Rb^+^	Cs^+^	Water Type	Place of Exploitation
				μS/cm	mg/L	µg/L	mg/L	mg/L	mg/L	mg/L	mg/L	mg/L	mg/L	mg/L	µg/L	µg/L	µg/L	µg/L		
1	Salutis	S	5.2	49	32	16	7.8	1.2	3.8	1.8	4.3	0.7	0.8	0.57	10	1	2.4	0.04	Na-Cl	Ferreira—Paredes de Coura
2	Fastio	S	6.0	35	141	19	3.8	8.5	2.3	0.8	4.3	0.6	1.2	0.47	1	<1	1.7	0.11	Na-HCO_3_	Chamoim—Terras de Bouro
3	Monchique	S	9.4	419	107	1058	31.7	126.9	nd	47.8	76.9	2.0	1.3	0.05	<1	1	5.1	0.03	Na-HCO_3_	Caldas de Monchique—Monchique
4	Vimeiro Lisa	S	7.1	83	75	<10	8.4	21.4	0.4	3.6	9.2	0.3	5.6	1.46	<1	1	0.3	0.02	Na-HCO_3_	Maceira—Torres
5	Vitalis	S	5.7	50	71	28	6.3	3.7	2.2	2.7	5.2	2.1	0.9	0.55	8	1	14.5	0.34	Na-Cl	Castelo de Vide
6	Caldas de Penacova	S	5.5	48	41	<10	7.8	4.3	1.8	1.3	6.1	0.3	0.7	1.10	4	2	0.9	0.19	Na-Cl	Penacova
7	Magnificat	NC	5.0	166	234	533	18.0	81.0	13.6	4.2	21.9	9.7	8.4	5.04	142	3	33.9	0.15	Na-HCO_3_	Serra do Trigo—Açores
8	Castello	AC	5.4	793	472	123	46.2	361.7	18.3	18.0	34.3	0.9	94.8	26.20	<1	7	0.4	0.04	Ca-HCO_3_	Pisões -Moura
9	Luso	S	5.6	58	56	31	7.3	12.2	1.5	1.4	7.4	0.8	0.8	1.81	3	7	3.5	0.39	Na-Cl	Luso—Mealhada
10	Vimeiro	AC	5.7	1050	2291	221	176.3	425.8	7.8	79.4	144.0	3.7	112	27.1	<1	11	3.4	0.19	Na/Ca-HCO_3_	Maceira—Torres
11	Carvalhelhos	S	7.0	248	453	963	2.6	141.5	0.2	6.9	55.0	1.4	5.6	0.70	1	173	16.3	30.90	Na-HCO_3_	Carvalhelhos -Boticas
12	Carvalhelhos	AC	5.3	189	208	498	2.7	124.4	1.1	7.6	52.3	1.4	5.9	0.62	<1	177	17.9	32.60	Na-HCO_3_	Carvalhelhos -Boticas
13	Melgaço	NC	5.7	840	439	657	10.6	691.7	1.2	7.5	87.5	3.6	145	3.01	275	600	19.5	6.73	Ca-HCO_3_	Quinta do Peso—Melgaço
14	Campilho	AC	5.9	1892	1289	4131	15.6	1288.3	0.9	8.7	428.0	27.1	37.4	10.06	1	1590	215.0	222.00	Na-HCO_3_	Vidago—Chaves
15	Frize	NC	6.5	2300	2336	1440	100.2	1941.0	1.6	nd	630.0	41.1	75.6	25.05	53	1760	335.0	331.00	Na-HCO_3_	Sampaio—Vila Flor
16	Pedras Salgadas	NC	6.1	2660	1825	1265	22.9	1897.1	0.2	7.5	594.0	34.4	95.6	24.70	213	1800	238.0	49.00	Na-HCO_3_	Pedras Salgadas—Vila Pouca de Aguiar
17	Bem-Saúde	NC	6.0	2310	1600	2100	90.0	1596.0	20.3	7.2	510.0	46.0	84.0	21.00	100	2000	na	na	Na-HCO_3_	Sampaio -Vila Flor
18	Vidago	NC	6.0	1554	1797	25	25.5	1869.0	0.8	7.7	624.0	53.8	73.7	14.40	20	2210	415.0	253.00	Na-HCO_3_	Vidago—Chaves

Notes: S—still natural mineral water; NC—naturally carbonated natural mineral water; AC—artificially carbonated natural mineral water; na—not available; nd—not detected.

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
