# Peer review of "Lithium in Portuguese Bottled Natural Mineral Waters—Potential for Health Benefits?"

_ijerph, 2020, doi:10.3390/ijerph17228369_

Round 1

Reviewer 1 Report

The manuscript presents the issues highlighted in the first round of revision 

Author Response

“The manuscript presents the issues highlighted in the first round of revision.”

authors´comment: not applicable

Reviewer 2 Report

In my opinion, the revised version of the manuscript can be published in its current form.

Author Response

Reviewer #2

“In my opinion, the revised version of the manuscript can be published in its current form.”

authors´ comment: not applicable

Reviewer 3 Report

The aim of the paper is to provide a mapping of concentration levels of lithium in bottled mineral waters wich could contribute to a dietary exposure assessment of the population. This subject is an important public health one, considering several diseases like bipolarity and Alzheimer to name only two. The present study relates Lithium concentrations to its hydrogeological contexte what makes it interesting in that this suggests a potential to predict geographic areas less at risk than others to be confronted to mental disorder. Such studies might thus have relevant impacts on public health management actions. 

Strenghts: The methodology was rigourously conducted. The subject is extensively documented and the review is interesting. Not only lithium was analyzed and reported but also elements like calcium and sodium which could be usefull to  assess the biological impact of lithium.

Weakness: The introduction could be structured a bit more to facilitate the reading and to keep the attention of the reader. For the same reason, the "Results and disccussion" paragraph could easly be splitted into "Results"  on the one hand and "Discussion" in the other hand : this would strcuture the paper into smaller paragraph from which the reasoning would be highlighted. See specific comments below for a suggestion.

Figure 2:The reader has to refer to the table and search by himself which samples are rich in lithium or not in the Piper diagram. I suggest double-purpose symbols in the legend for the Piper diagram: the ones used reflects the composition of water only.  For example, sample 13 (high Li) is a black triangle such as sample 8 (low Li); sample 10 is high Li and has the same symbol exactly than samples 2,3,4,7;... This is scientifically correct but to help the reading of the graph, I suggest to use empty triangles for low Li Ca-HCO3 waters and empty squares for low Li NaHCO3 waters, and filled triangles for high Li HCO3 waters and full squares for high Li NaHCO3 waters.

Subtitle Discussion at the rows 253 and 294; Results from row 239-252 and from row 280-293.

The introduction is well documented and interesting but it is quite long, so to keep the attraction of the reader, I would suggest the following. The challenge is the improvement of human health thus, I suggest to first introduce what do we know about therapeutic use of lithium [row 35-54 synthesized with 89-94 synthesized with 98-102+95-97] and then about dietary as a source of Li exposure in general [row 66-77]  with then an emphasis on drinks and especially mineral waters [103-106]. Then, I would introduce the legislation aspects for bottled waters [107-142] after having extracted the paragraph about Portuguese legislation [119-124+136-142] to keep it for the end of this sub-paragraph. Then I would introduce the link between water composition and the hydrogeochemical context [55-58+103-105] in general. This last point justifies the approach of the study [143-149] and links it smootly with the public health issue.

row 37-38:"but the clinical…, but which proved…" please rephrase to avoid the repetition

row 43:receptor(s)  …, increasing (serotonin)…

Author Response

Reviewer 3

Comments and Suggestions for Authors

The introduction could be structured a bit more to facilitate the reading and to keep the attention of the reader. For the same reason, the "Results and discussion" paragraph could easily be splitted into "Results" on the one hand and "Discussion" in the other hand: this would strcuture the paper into smaller paragraph from which the reasoning would be highlighted. See specific comments below for a suggestion.

Authors´ comment:

The Reviewer is suggesting a considerable restructuring of the manuscript. It is important to note that the manuscript in its present form is already the product of a previous restructuring suggested by previous reviewers #1, #2 and #4. This restructuring carried out was appreciated, as can be easily deduced from the comments presented:

Reviewer #1

“The manuscript presents the issues highlighted in the first round of revision.”

Reviewer #2

“In my opinion, the revised version of the manuscript can be published in its current form.”

Reviewer #4

“Clearly the manuscript has been though significant review and revision as evidenced by the red underline and yellow highlights.  … but the content is strong and will be of interest to readers.”

R3.1: Subtitle Discussion at the rows 253 and 294; Results from row 239-252 and from row 280-293.

R3.2: The introduction is well documented and interesting but it is quite long, so to keep the attraction of the reader, I would suggest the following. The challenge is the improvement of human health thus, I suggest to first introduce what do we know about therapeutic use of lithium [row 35-54 synthesized with 89-94 synthesized with 98-102+95-97] and then about dietary as a source of Li exposure in general [row 66-77]  with then an emphasis on drinks and especially mineral waters [103-106]. Then, I would introduce the legislation aspects for bottled waters [107-142] after having extracted the paragraph about Portuguese legislation [119-124+136-142] to keep it for the end of this sub-paragraph. Then I would introduce the link between water composition and the hydrogeochemical context [55-58+103-105] in general. This last point justifies the approach of the study [143-149] and links it smootly with the public health issue.

Authors´ comment:

The Reviewer´s comments and suggestions highlight that we should reorganize the text of the Introduction and split the Results and Discussion section, based on the order presented by the reviewer. An attempt was made to follow these suggestions but the final result did not improve the sequence in order to contribute to a better linkage of the information available to the reader.

In fact, we think that there are several models for writing the Introduction to a scientific work, and that the model suggested by reviewer # 3 could be one of them. However, this Introduction has already been rewritten according to the suggestions of previous reviewers #1, #2 and #4. From our point of view, as it is, the present Introduction describes why we are writing this paper and supplies sufficient background information for the reader to understand and evaluate the rational of the manuscript.

In our view, the suggested changes are not like-minded with the “Minor Revisions” indicated by the Editor.

However, other recommendations of the reviewer were considered as requested.

Authors agree with the suggestion to extract the paragraph related with Portuguese legislation (119-124) and move it to other place of the text that was focus mainly in the Portuguese natural mineral water (see Line 141-146).

R3.3: row 37-38:"but the clinical…, but which proved…" please rephrase to avoid the repetition….  and row 43:receptor(s)  …, increasing (serotonin)…

Authors´ comment:

Both corrections were performed “as requested” by the reviewer.

R3.4: Figure 2:The reader has to refer to the table and search by himself which samples are rich in lithium or not in the Piper diagram. I suggest double-purpose symbols in the legend for the Piper diagram: the ones used reflects the composition of water only.  For example, sample 13 (high Li) is a black triangle such as sample 8 (low Li); sample 10 is high Li and has the same symbol exactly than samples 2,3,4,7;... This is scientifically correct but to help the reading of the graph, I suggest to use empty triangles for low Li Ca-HCO3 waters and empty squares for low Li NaHCO3 waters, and filled triangles for high Li HCO3 waters and full squares for high Li NaHCO3 waters.

Authors´ comment:

The authors´ agree with the comments to improve the reading of Figure 2 and performed the suggested corrections in Piper Diagram legend and in the Figure 2 caption (see Lines 265-268)

Reviewer 4 Report

Clearly the manuscript has been though significant review and revision as evidenced by the red underline and yellow highlights. At this point there remains a few typographical errors (e.g., L260, L271, L361, etc.) but the content is strong and will be of interest to readers.

The line about causality (273 274 red and highlighted should be deleted). Association does not imply causality and this topic is too complex for this paper. Change the sentence to read: Concerning the protective effect of low exposure to Li from drinking water, future epidemiologic studies are warranted.

Author Response

Reviewer 4

“Clearly the manuscript has been though significant review and revision as evidenced by the red underline and yellow highlights.  … but the content is strong and will be of interest to readers.”

R4.1: There remains a few typographical errors (e.g., L260, L271, L361, etc.)

Authors´ comment:

These and other corrections were performed as suggested by the reviewer.

R4.2: The line about causality (273 - 274 red and highlighted should be deleted). Association does not imply causality and this topic is too complex for this paper. Change the sentence to read: Concerning the protective effect of low exposure to Li from drinking water, future epidemiologic studies are warranted.

Authors´ comment:

Both corrections and suggestion were performed (see Lines 305-306).

This manuscript is a resubmission of an earlier submission. The following is a list of the peer review reports and author responses from that submission.

Round 1

Reviewer 1 Report

The ijerph-901288 manuscript reports a study on the Portuguese quality of bottled water. The study is only descriptive, there is no data processing. the manuscript lacks originality. Moreover, I think the title of the manuscript, the abstract and the introduction are a bit speculative because they describe a series of events and effects hypothesized and not analysed in the present study.

Especially the Title and lines 296-302 may give the wrong or at least unproven impression.

There is no reliable evidence that lithium is effective in treating mental illness. On the contrary, it is known that high concentrations of metals are toxic to human health.

I strongly suggest that the authors change the title of the manuscript and emphasize that there is no evidence of the actual use of lithium as a drug. I suggest that the authors send the manuscript to a local or international magazine that focuses on water monitoring

Author Response

"Please see the attachement"

Reviewer 2 Report

The work presented to me for review entitled "Lithium, the Magic Ion on Portuguese Bottled Natural Mineral Waters" is suitable for publication after the improvements described below.

The title should be formulated in a more scientific way to make it easier to search for in databases later on.

The introduction is a bit too detailed and finicky. The authors report that concentrations of lithium ions in the therapy of mental disorders are higher than those available in mineral waters (lines 92-93), and at the same time they refer to works indicating the correlation between lithium consumption and the number of suicides (lines 74-84) or neurodegenerative diseases (lines 85-86). It would therefore be appropriate to include in the introduction information on the possibility of this element accumulating in the body or to clarify the role of regular intake with drinking water.

The material and methods section lacks an explanation of the described division into low and high lithium waters. The information about adopted values should have been added.  

Tere isn’t any information on the working conditions for the inductively coupled plasma mass spectrometer (ICP-MS) and on the LOD and LOQ values for the individual elements.

It also lacks information on the packaging in which the analysed waters were stored (plastic or glass bottle). The type of packaging can affect the content of trace elements, so this information should be included in the M&M section.

In my opinion the phrase "natural mineral" from Table 1 should be changed to "natural ions" and their charge should be given in the upper index.

The description of the results and the discussion are written correctly, although in my opinion a little too laconic. It is worth extending them using the latest scientific reports.

References should be carefully checked because they contain formatting errors and repetitions, as for example items 3,48,63 are the same article.

Reviewer 3 Report

Comments:

  1. row 103, Shouldn't it be "whwre the environment is NOT well protected?
  2. 123: delete one "for"
  3. 226: "is this group" should be "in this group"
  4. 272-273: Ca-carbonated also helps neutarlize acidity, write down. 
  5. 284: According TO Calado...
  6. 297: Good, all minerals in water are more readily absorbed in the intestines. In Rosborg et al 2003, Hair element concentrations... There was a significant relaton between Ca in well water and hair, as well as other minerals. 
  7. 318: Not modified, but modify
  8. 335: Vidago has too high Na-concentration to be recommended as a source for Li. Sorry. 
  9. But the article is very good, except for that. It's needed.....
